# Osteoblast-Derived Matrix Vesicles Exhibit Exosomal Traits and a Unique Subset of microRNA: Their Caveolae-Dependent Endocytosis Results in Reduced Osteogenic Differentiation

**DOI:** 10.3390/ijms241612770

**Published:** 2023-08-14

**Authors:** Anne M. Skelton, D. Joshua Cohen, Barbara D. Boyan, Zvi Schwartz

**Affiliations:** 1Department of Physiology and Biophysics, School of Medicine, Virginia Commonwealth University, Richmond, VA 23298, USA; skeltonam@vcu.edu (A.M.S.); bboyan@vcu.edu (B.D.B.); 2Department of Biomedical Engineering, College of Engineering, Virginia Commonwealth University, Richmond, VA 23284, USA; djcohen@vcu.edu; 3Wallace H. Coulter Department of Biomedical Engineering, Georgia Institute of Technology, Atlanta, GA 30332, USA; 4Department of Periodontics, University of Texas Health Science Center at San Antonio, San Antonio, TX 78229, USA

**Keywords:** matrix vesicles, extracellular vesicles, osteoblasts, osteogenesis, endocytosis, differentiation, microRNA, caveolin

## Abstract

Matrix vesicles (MVs) are nano-sized extracellular vesicles that are anchored in the extracellular matrix (ECM). In addition to playing a role in biomineralization, osteoblast-derived MVs were recently suggested to have regulatory duties. The aims of this study were to establish the characteristics of osteoblast-derived MVs in the context of extracellular vesicles like exosomes, assess their role in modulating osteoblast differentiation, and examine their mechanism of uptake. MVs were isolated from the ECM of MG63 human osteoblast-like cell cultures and characterized via enzyme activity, transmission electron microscopy, nanoparticle tracking analysis, Western blot, and small RNA sequencing. Osteoblasts were treated with MVs from two different culture conditions (growth media [GM]; osteogenic media [OM]) to evaluate their effects on the differentiation and production of inflammatory markers and on macrophage polarization. MV endocytosis was assessed using a lipophilic, fluorescent dye and confocal microscopy with the role of caveolae determined using methyl-β-cyclodextrin. MVs exhibited a four-fold enrichment in alkaline phosphatase specific activity compared to plasma membranes; were 50–150 nm in diameter; possessed exosomal markers CD63, CD81, and CD9 and endosomal markers ALIX, TSG101, and HSP70; and were selectively enriched in microRNA linked to an anti-osteogenic effect and to M2 macrophage polarization. Treatment with GM or OM MVs decreased osteoblast differentiation. Osteoblasts endocytosed MVs using a mechanism that involves caveolae. These results support the hypothesis that osteoblasts produce MVs that participate in the regulation of osteogenesis.

## 1. Introduction

Populations of various heterogenous, nano-sized, membrane-bound extracellular vesicles are increasingly becoming the focus of regenerative medicine as they contain bioactive lipids, proteins, and nucleic acids [1]. While some populations of these vesicles are found in biological fluids, similar vesicles, termed matrix vesicles (MVs), are also present in the extracellular matrix (ECM) of mineralizing tissues like growth plate cartilage, bone, and dentine. MVs are released by chondrocytes and osteoblasts to become anchored in the ECM via integrin binding to collagen fibrils [2,3,4,5].

Historically, MVs were identified as sites of initial hydroxyapatite formation during endochondral ossification, varying in phospholipid composition and matrix processing enzymes as a function of chondrocyte maturation in the growth plate [3,6,7]. Their composition is regulated genomically, and they exhibit functions independent of calcium phosphate crystal nucleation, suggesting that they are involved in growth plate regulation in ways other than mineral deposition [8,9]. For example, matrix metalloproteinases present in growth plate chondrocyte MVs can activate latent transforming growth factor-β (TGF-β) found in the ECM in a 1,25-dihydroxy vitamin D_3_-dependent manner [10].

Recent studies found the presence of specific microRNA (miRNA) in MVs produced by growth plate chondrocytes, indicating selective packaging of miRNA cargo into MVs by parent cells. miRNA are short, non-coding RNA approximately 19–25 nucleotides in length that regulate target messenger RNA (mRNA) [11,12]. Sequence complementarity allows one miRNA to target up to hundreds of mRNA, leading to their involvement in the pathogenesis of many disease states, as well as normal physiology [11,12]. MVs from different maturation levels of the growth plate exhibited differences in miRNA enrichment and those differentially upregulated miRNA were found to regulate cell proliferation and maintain an extracellular environment that supported those cellular phenotypes [4,13,14].

Less is known about MVs that are present in bone. Studies examining MVs produced by osteoblast-lineage cells in culture show they are not identical to MVs produced by chondrocytes, especially with regard to enzyme enrichment and responsiveness to hormones [3,6,15]. The role of MVs as initial sites of mineralization in the growth plate has been well described, as has initial crystal formation in MVs present in primary bone and osteoid. It is less clear whether MVs play a similar role in mature bone, where nucleation sites exist within the collagen fibrils [2,16,17,18]. Moreover, the heterogeneity in isolated MVs with respect to size and composition suggests that there are different species of MVs with distinct functions in the tissue.

Osteoblast-lineage cells, including perivascular stem cells, bone marrow stromal cells (MSCs), osteoblast-like cell lines, and human osteoblasts, also produce classical exosomes, which are a class of extracellular vesicles that are not anchored to the ECM. The treatment of MSC cultures with these exosomes upregulated osteogenic markers and enhanced matrix mineralization [19,20,21], supporting the hypothesis that extracellular vesicles from osteoblast-lineage cells can regulate cell-to-cell communication and can affect osteogenic differentiation in target cells [22,23]. These observations suggest that MVs, which are similar to exosomes in many respects, may also play a regulatory role in bone.

The goal of the current study was to characterize MVs produced by osteoblasts and evaluate their role and mechanism of action in bone. The proposed study adds to the growing body of literature regarding extracellular vesicles including specific MV characteristics when derived from an osteoblast-like cell, their impact on osteoblast differentiation and inflammation, and their mechanism of uptake by target cells. We initially hypothesized that MVs would mediate osteogenesis by increasing markers of osteoblastic differentiation similar to exosomes. MVs produced by MG63 cells were found to exhibit the characteristic size and morphology of exosomes but are distinct from exosomes produced by the same cells. Their unique miRNA cargo contributed to a role in which they significantly decrease osteoblastic differentiation markers, regardless of their culture conditions, via caveolae-mediated endocytosis, while at the same time promoting the polarization of human macrophages to a pro-regenerative M2 phenotype.

## 2. Results

### 2.1. Matrix Vesicles Display Exosomal Characteristics

MVs were isolated at 24 h post-confluence from the trypsinized ECM of human osteoblast-like MG63 cells that were cultured in growth media. After pelleting the cells, cells were lysed and plasma membranes (PM) isolated via centrifugation with a sucrose gradient. Alkaline phosphatase specific activity of the MV fraction was 5-fold higher than the PM fraction, validating the purity of the MV isolation (Figure 1A). Transmission electron microscopy confirmed a heterogenous population of vesicles in the MV isolate with a circular morphology and bilaminar membrane (Figure 1B,C). Size distribution was further confirmed by nanoparticle tracking analysis, demonstrating a median size of 105 nm and a mean size of 115 nm with a distribution between 30 and 200 nm (standard deviation ± 85 nm; Figure 1D). Western blots of the cell pellet (CP), PM, and MV fractions showed that the tetraspanins CD63, CD9, and CD81, which are classical markers of exosomes, were found only in the MV fraction. Endosomal pathway markers ALIX, HSP70, and TSG101 were present in the CP, PM, and MV fractions, but relative intensities differed depending on the fraction. ALIX and TSG101 were most enriched in the MV fraction with HSP70 displaying the greatest intensity in the CP fraction. Na^+^-K^+^ ATPase and annexin V, two proteins responsible for maintaining the calcium gradient within MVs along with other cellular roles, were found in all three fractions as well (Figure 1E). Whereas Na^+^-K^+^ ATPase was enriched in the PM and MV fractions, annexin V was enriched in the CP fraction. GAPDH served as the internal control for all fractions.

### 2.2. MVs Are Endocytosed by MG63 Cells

The confocal microscopy of PKH26-labeled MVs added to counter-stained MG63 cells (Figure 1F,G) demonstrated the visible uptake of vesicles by cells compared to cells incubated with the 0.9% NaCl vehicle control (Figure 1H). When quantified by uptake percentage, 44% of MV-treated cells had at least one fluorescent spot compared to 15% of NaCl-treated cells, constituting a significant difference (Figure 1I).

### 2.3. microRNAs Are Selectively Packaged into MG63 MVs

MV and CP fractions underwent small RNA isolation and sequencing followed by differential expression analysis. Principal component analysis, a measure of variability of the data set with reduced dimensionality, demonstrated distinct grouping of the CP and MV fractions (Figure 2A). The further apart the samples are from one another, the more significance that can be expected in the expression of miRNA. The results indicate that the CP and MV fractions have distinct populations of miRNA with slightly greater variability between samples in the MV group as the samples are more spread apart on the plot. The enriched miRNA, that is those with an absolute Log2 fold change greater than 2 (>2), in the MVs and CPs were then evaluated for their unique expression based on raw read counts. Ninety-six of the differentially expressed miRNA were found exclusively in MVs while 168 were found exclusively in the CP fraction. A total of 229 had an increased Log2 fold change between the two fractions but still had some level of raw read counts in the other fraction (Figure 2B). While this diagram does not differentiate between those differentially expressed microRNAs that are up- or downregulated, it does demonstrate the specific enrichment in MVs by parent cells as there are miRNAs found exclusively in the MV fraction.

### 2.4. Differentially Expressed MV miRNA Indicate Regulatory Roles in Osteoblasts and a Regenerative Role in Macrophages

The differentially expressed miRNA were further characterized, demonstrating the unique populations between MVs and CPs, shown in the heatmap exhibiting clusters based on the similar expression patterns of miRNA (Figure 2C). A volcano plot of those miRNAs with a Log2 fold change > 2 and an adjusted *p*-value less than 0.05 (<0.05) in the MV samples, demonstrated that hsa-miR-122-5p, hsa-miR-451a, and hsa-miR-144-3p had the greatest fold change compared to the CP fraction (Figure 2D). Fourteen miRNAs that were considered the most differentially expressed were further explored in the literature. Nine were linked to anti-osteogenic pathways, found to decrease differentiation and mineralization in vitro, or decrease bone formation in vivo. In contrast, three of the miRNAs were pro-osteogenic, increasing differentiation and mineralization in vitro or increasing bone formation in vivo. Two of the miRNAs, miR-122-5p and miR-451a, were deemed inconclusive as there was published evidence for both pro-osteogenic and anti-osteogenic potential (Table 1, Figure 3A). Additionally, fourteen miRNAs were explored in the literature for any effect on macrophage polarization. Four of the miRNAs were found to promote polarization to an M1 phenotype, associated with the acute inflammatory response to injury or infection. Ten of the miRNAs were found to promote polarization to an M2 phenotype, which is considered the regenerative type of macrophage that promotes an inflammatory environment conducive to cellular turnover and growth (Table 2, Figure 3B).

### 2.5. Treatment of MG63 Cells with Matrix Vesicles Decreases Markers of Osteoblast Differentiation

The treatment of MG63 cells with MVs did not alter the DNA content of the cultures (Figure 4A). However, the addition of MVs to confluent cultures for 24 or 48 h caused a decrease in alkaline phosphatase specific activity, an early marker of osteoblast differentiation, and a decrease in osteocalcin production at 48 h (Figure 4B,C). Osteopontin production was not affected by treatment with MVs (Figure 4D), but there was a significant decrease in the production of vascular endothelial growth factor (VEGF), a stimulator of blood vessel formation (Figure 4E). Interleukin-6 (IL-6), a cytokine promoting chronic inflammation, was significantly decreased at 24 and 48 h (Figure 4F), whereas the regenerative inflammatory cytokine interleukin-10 (IL-10) was not affected (Figure 4G). When mRNA was quantified, BGLAP mRNA did not show any differences with MV treatment at either 24 or 48 h (Figure 4H). RUNX2 exhibited no difference between groups at 24 h but was significantly increased with MV treatment at 48 h (Figure 4I). In contrast, osterix/SP7 was decreased with MV treatment at 24 h; no difference between groups was observed at 48 h (Figure 4J).

### 2.6. MV Composition Is Sensitive to Culture Conditions

MVs were isolated from MG63 cells cultured for 10 days in osteogenic media (OM) containing ascorbic acid, β-glycerophosphate, and dexamethasone. The treatment of naive cultures with these MVs resulted in a slight decrease in DNA content at 48 h (Figure 5A). Alkaline phosphatase-specific activity was reduced at 48 h and osteocalcin production was reduced at both 24 and 48 h of treatment (Figure 5B,C). MVs from the cultures grown in OM had no effect on production of either osteopontin (Figure 5D) or VEGF (Figure 5E). IL-6 was reduced at 24 and 48 h (Figure 5F), whereas IL-10 was significantly increased at 48 h (Figure 5G).

### 2.7. Exosomes Display Characteristics Different from Matrix Vesicles

Exosomes (EXs) were isolated from the conditioned media and characterized as described for MVs above. Alkaline phosphatase-specific activity was significantly greater than that of the cell pellet (CP) fraction but at 50 nmol Pi/mg protein/minute, was lower than observed in MVs (Figure 6A). The size distribution of EXs, as determined by nanoparticle tracking analysis, showed that they were smaller than MVs with an average size of 59 nm (±44.6 nm standard deviation) and a median size of 42 nm (Figure 6B). Western blots showed that CD9 and CD81 were present only in the EX fraction as compared to the CP. ALIX was greater in the EX fraction while HSP70 was greater in the CP. Na-K ATPase, TSG101, and annexin V were equally distributed between the CP and EXs (Figure 6C).

### 2.8. MG63 Cells Respond Differentially to MVs and EXs

MG63 osteoblast-like cells were treated for 48 h with 5 µg/mL (protein concentration) of either MVs or EXs isolated from the trypsinized cell layer or conditioned media, respectively, of MG63 cells cultured in growth media. At harvest, the treated cells were assayed for their DNA content, alkaline phosphatase-specific activity of the cell layer, and osteocalcin content of the conditioned media. No significant differences were found in DNA (Figure 6D). MVs caused significant decreases in both alkaline phosphatase specific activity and osteocalcin while EXs had no effect on enzyme activity or osteocalcin production (Figure 6E,F).

### 2.9. Inhibition of Caveolae-Mediated Endocytosis Prevents Uptake and Effect of MVs on Osteoblast-Like Cells

MG63 osteoblast-like cells were treated with methyl-β-cyclodextrin (β-CD), a chemical destroyer of cholesterol in the plasma membrane, to inhibit caveolae-mediated endocytosis, followed by treatment with MVs isolated from the trypsinized cell layer of MG63 cells cultured in growth media. MV uptake was visualized using confocal microscopy. PKH26-labeled MVs were added to counter-stained MG63 cells pre-treated with vehicle or 10 mM β-CD; significantly more MV fluorescence was observed in the control cells (Figure 7A) than in the β-CD treated cells (Figure 7B). Morphometric analysis confirmed these results, and demonstrated MV uptake in 57% of the control cells compared to 1.5% of the β-CD-treated cells (Figure 7C).

Pre-treatment of the cultures with increasing concentrations of β-CD did not alter the DNA content of the cultures when the cells were treated with vehicle alone. In contrast, at the highest dose of β-CD (10 mM), MV treatment caused a small decrease in DNA (Figure 7D). MVs caused a reduction in alkaline phosphatase-specific activity in the vehicle-treated group, but this effect was lost when the cultures were pre-treated with β-CD (Figure 7E). Together, these results suggest that the disruption of the caveolae using β-CD inhibits the uptake of MVs and their downstream effects on osteoblast differentiation.

## 3. Discussion

To date, three subtypes of extracellular vesicles have been classified based on their biogenesis pathways: exosomes, microvesicles, and apoptotic bodies. Exosomes are the most extensively studied with biogenesis in the endosomal pathway prior to release into biological fluids [1]. Our results support the hypothesis that osteoblast-derived MVs are a class of exosomes, similar to other matrix-associated matrisomes. Electron microscopy demonstrated a circular morphology with a bilaminar membrane and a size distribution typical of exosomes: 50–150 nm and 30–150 nm, respectively [1,73]. MVs and exosomes share the enrichment of sphingomyelin and annexin A2 in their lipid and protein composition [73]. Western blots of the osteoblast-derived MVs demonstrated classical exosome markers, tetraspanins CD63, CD9, and CD81, confirming isolation of an extracellular vesicle population from the trypsinized ECM. Markers of endosomal biogenesis, HSP70, ALIX, and TSG101, were also found in the MV fraction, indicating that their production involved a similar mechanism to that used to produce exosomes. This finding is supported by earlier studies using somatic cell hybrids to demonstrate the specific genesis of MVs as distinct organelles and not due to the non-specific budding from the PM [74].

However, it is clear that like MVs isolated from growth plate cartilage, they are distinct from exosomes present in biological fluids. MVs exhibit marked enrichment in alkaline phosphatase-specific activity compared to the PMs isolated from their parent osteoblasts. Consistent with previously described MVs, they consist of a heterogenous population between 50–150 nm in size [14,22,75,76]. Earlier studies comparing membrane-bound vesicles in the media of chondrocyte cultures to the MVs isolated from the ECM indicate that the two populations are distinctly different, with a different lipid composition of their bilayers and different enzyme content. Importantly, the media vesicles did not exhibit an enrichment in alkaline phosphatase specific activity relative to the PM. MVs are tethered to the ECM of mineralizing tissues via integrins, and integrin-tethered exosomes have been reported in other tissues, primarily cancer, suggesting that osteoblast-derived MVs may share similarities with this class of matrisomes [77].

Similar to chondrocyte-derived MVs, those derived from the MG63 osteoblast-like cell cultures contained small, non-coding miRNAs. MVs derived from MC3T3-E1 cell cultures also contain miRNAs and microarray analysis showed that the MC3T3-E1 MVs differed from miRNA identified in chondrocyte- and MSC-derived MVs [22,78]. Our use of small RNA sequencing has a higher specificity and sensitivity than microarray, allowing the detection of a higher percentage of differentially expressed genes [79]. We found that the MG63 MVs contained uniquely differentially expressed miRNA from their parent osteoblast-like cells and this miRNA was selectively packaged into MVs, as demonstrated by the heatmap and volcano plot, which both show a distinct pattern of miRNA expression. While the difference in species and degree of osteoblastic differentiation makes a direct comparison difficult, our dataset and that of Minamizaki et al., have at least 11 similarly expressed miRNA of the 52 we found to be significantly upregulated in osteoblast-derived MVs [78]. It should also be noted that our MG63 cells and their MC3T3-E1 cells represent different levels of osteoblastic differentiation; therefore, similar to chondrocyte-derived MVs, there may be different expression levels of miRNA depending on parent cell maturity, explaining the 41 possibly unique miRNA [13,78].

When several miRNAs that were significantly upregulated in MG63 cell MVs were further explored in the literature, the majority of them were found to be anti-osteogenic; that is, they are linked to known anti-osteogenic pathways, found to decrease osteoblastic differentiation and mineralization in vitro, or decrease bone formation in vivo. While three miRNA were pro-osteogenic (promoted differentiation and/or mineralization) and two had conflicting evidence in the literature, the differential expression of miRNA in MVs appears to point toward a role in inhibition of osteoblastic differentiation and slowing of osteogenesis.

Further exploration of these miRNA in the literature revealed potential pathways of anti-osteogenic action including connexin 43, MAP kinase, and Wnt. miR-144-3p and miR-206 have both been linked to connexin 43, which is a vital signaling protein for skeletal development and maintenance [36,41,80]. miR-433-3p was found linked to five targets involved in the parathyroid hormone, MAP kinase, Wnt, and glucocorticoid signaling pathways, suggesting these miRNAs may function in multiple pathways as both activators and suppressors to produce anti-osteogenic effects [48].

This anti-osteogenic effect of MVs was corroborated by the in vitro treatment data with MVs grown in growth medium, as two markers of osteoblastic differentiation, alkaline phosphatase and osteocalcin, were decreased with MV treatment. Mizukami et al., found similar results when they treated ST2 osteoblast-like cells and primary osteoblasts in vitro with collagenase-released extracellular vesicles from the cell layer of mineralizing osteoblasts and observed decreased messenger RNA expression for osterix and osteocalcin [81]. MVs that were isolated from cultures of MG63 cells grown in osteogenic medium for a longer period of time also decreased osteoblastic differentiation markers, which suggests the anti-osteogenic role is inherent to the MVs themselves. When mRNA expression was evaluated after treatment with MVs that were isolated from GM cultures, only osterix was found to be significantly decreased at the 24 h time point. However, mRNA expression may be dependent on many factors, including the condition of the cells, the treatment timeline, and cell harvest conditions, which is why validation with ELISA protein expression is helpful to understand the complete picture of the effect of MVs after 24 and 48 h of treatment.

When compared to the role of exosomes, which are extracellular vesicles released into culture medium, MVs appear to serve a different role in bone. Exosomes from mineralizing osteoblast-like cells, human osteoblasts, and perivascular stem cells upregulated alkaline phosphatase, osteocalcin, and RUNX2 in vitro and enhanced matrix mineralization within MSC cultures as well as improving bone regeneration and healing speed of critical sized calvarial defects in vivo [19,20,21,82]. Our characterization data, particularly, the presence of microsomal markers evident in the western blots, suggest that MVs and exosomes have a similar biogenesis pathway, size, and morphology. However, it appears that their location within mature bone, anchored to the ECM versus free-floating, does dictate their role.

Direct comparison of the treatment of MG63 cells with either MVs or exosomes from MG63s cultured in growth medium, clearly demonstrates differences in their effects on alkaline phosphatase and osteocalcin. While these markers did not increase with exosome treatment, they also did not decrease as seen with MV treatment. This suggests that MVs and EXs are separate types of extracellular vesicles with different roles when produced by osteoblasts. Exosomes may communicate with more distant cells to coordinate osteogenic efforts, while MVs remain close to parent cells to modulate the rate of osteogenesis. Further exploration of MVs compared to exosomes is necessary to resolve this dichotomy, including increasing the treatment timeline to possibly establish the different roles of these extracellular vesicles.

The role of MVs becomes even more interesting when exploring the significantly upregulated miRNA with respect to macrophage polarization. A significant number of the MV miRNA were linked to M2 polarization and this is corroborated by the in vitro data that demonstrated significant decreases in IL-6 production by osteoblasts treated with MVs and an additional significant increase in IL-10 production caused by MVs from cultures grown in osteogenic medium. IL-6 and IL-10 are two cytokines associated with M1 and M2 macrophages, respectively, and these two pieces of data suggest that osteoblast-derived MVs promote M2 polarization, or the regenerative phenotype of macrophages, over M1 polarization [83]. MVs, while they may not directly promote the osteogenic differentiation of osteoblasts or the production of factors needed for bone formation, still do promote a regenerative environment within bone and may ultimately contribute to the formation of new bone in a more nuanced and complicated manner. This hypothesis is supported by Mizukami, et al., who found increased bone healing and increased chondrogenesis of a femoral defect treated with collagenase-released extracellular vesicles from the cell layer of mineralizing osteoblasts in vivo [81].

Studies using PKH26, a lipophilic, fluorescent dye that stains the membrane of the vesicles, demonstrated that MVs were endocytosed by the MG63 cells. Confocal microscopy of cells pretreated with methyl-β-cyclodextrin, which depletes cholesterol to inhibit caveolae-mediated endocytosis, had a visibly significant decrease in fluorescence compared to cells treated with vehicle. Furthermore, β-CD-treated cells did not exhibit the decrease in alkaline phosphatase seen with MV treatment. This suggests that MVs are organelles that are endocytosed in a caveolae-dependent manner. Other mechanisms for MV uptake may play a role, particularly if more than one species of MV is present in the heterogeneous population isolated from the ECM. Consensus regarding cellular uptake of extracellular vesicles remains to be reached as some research has identified the involvement of caveolin [84,85], while others support a role of caveolin-independent endocytosis [86,87], such as clathrin-dependent endocytosis [88]. However, the loss of the inhibitory effect on cellular alkaline phosphatase elicited by MVs in the β-CD treated cultures indicates that caveolae are involved.

This study took advantage of a well-characterized cell line, which has been used extensively to study the regulation of osteoblast differentiation by orthopaedic and dental implant topography [89,90]. MG63 human osteoblast-like osteosarcoma cells were originally isolated from a male patient [91] and have been shown to have characteristics typical of immature osteoblasts [92]. As a result, conclusions as to the role of MVs in regulating mature osteoblasts are limited. However, the recent literature continues to support their use as a model for phenotypic changes in osteoblasts, even demonstrating their ability to mineralize the ECM [93,94]. We selected these cells because they enabled us to look at early differentiation events and to assess the potential interaction of cells in the osteoblast lineage with immunomodulatory cells that would be in the same microenvironment. We believe the mRNA expression of osterix, RUNX2, and osteocalcin by these cells with and without MV treatment demonstrates their phenotype as an osteoblast and supports their use in this study. It is certainly possible that the MVs produced by these cells contribute to the proliferation of osteosarcoma cells, in addition to providing an internal brake on too rapid synthesis and mineralization of osteoid. Future studies using normal human osteoblasts will provide additional insights into the mechanism of action of MVs in bone, especially with regard to different stages of osteoblastic differentiation, longer treatment with MVs, and effects on mineralization of the ECM.

In summary, osteoblast-derived MVs share many characteristics with MVs from other mineralizing tissues like growth plate cartilage, as well as with extracellular vesicles like cancer-derived exosomes, but they also exhibit properties distinct from exosomes (Figure 8). Our small RNA sequencing demonstrated the enrichment of miRNA within the osteoblast-derived MVs, specifically those that inhibit osteoblast differentiation and promote a regenerative inflammatory state. The results suggest that osteoblast-derived MVs may modulate the rate of osteogenesis as well as contribute in a much more nuanced manner involving M2 polarization of macrophages to a regenerative phenotype. They appear to exert this effect via caveolae-mediated endocytosis. Further studies will continue to explore the impact of MVs on cells within mature bone, their uptake mechanisms to exert these effects, and the possible therapeutic application of MVs.

## 4. Materials and Methods

### 4.1. Osteoblast Cultures

MG63 cells were plated in T-175 flasks at a density of 10,000 cells/cm^2^ in Dulbecco’s modification of Eagle’s medium (DMEM) containing 1.0 g/L glucose (Corning Inc., Corning, NY, USA), 10% fetal bovine serum (FBS; Gemini, West Sacramento, CA, USA), and 100 U/mL penicillin and streptomycin (pen-strep). Cells were cultured at 37 °C in 5% CO_2_ with media changes 24 h after plating and every 48 h thereafter. Cells reached confluence after approximately four days. For matrix vesicle and exosome isolation, MG63 cells were cultured in DMEM full media and were harvested as described below 24 h after reaching confluence (growth media). Alternatively, MG63 cells were cultured in DMEM full media supplemented with 10 mM β-glycerophosphate, 50 µg/mL ascorbic acid, and 10^−8^ M dexamethasone for 10 days (osteogenic media).

### 4.2. Preparation of Exosome-Depleted Culture Medium

FBS was added to basal DMEM to a final concentration of 20% FBS and mixed gently. The medium was transferred to ultracentrifuge tubes, balanced, and centrifuged at 100,000× *g* for 16 h at 4 °C (33,000 RPM in 50.2 Ti Rotor using a Beckman Coulter Class R ultracentrifuge). Following ultracentrifugation, the supernatant was transferred to be sterile filtered while not disturbing the pellet and discarding the final 2–3 mL of liquid in each tube. The pooled supernatant was filtered with 0.22 µm filter in the cell culture hood and stored at 4 °C until use. At the time of use, the exosome-depleted media was diluted with an equal volume of DMEM and pen-strep was added to the appropriate concentration. Exosome-depleted media were added to each flask 24 h prior to harvest after washing the cell layer twice with sterile DPBS.

### 4.3. Cell Harvest and Isolation of Plasma Membrane, Matrix Vesicle, and Exosome Fractions

At harvest, the culture media from each flask of cells were pipetted and collected. The cell layer, including the extracellular matrix and cells, was washed twice with 10 mL of sterile DPBS. The cell layer was then trypsinized with 10 mL of sterile 0.25% trypsin and placed in 37 °C incubator for four minutes. After four minutes, the cell layer was scraped with a cell scraper and placed back in the incubator for a total time of 10 min. The trypsin in each flask was quenched with 10 mL of sterile DPBS containing 10% exosome-free FBS. To obtain exosome-free FBS, complete FBS was ultracentrifuged at 180,000× *g* for 4 h at 4 °C (44,500 RPM in 50.2 Ti rotor using a Beckman Coulter Class R ultracentrifuge). Following ultracentrifugation, the top ½ of the FBS in each tube was selected, leaving behind the pellet and cloudier supernatant, and sterile filtered. Exosome-free FBS was stored at −20 °C until the day of harvest.

Following quenching of the trypsin, cells were separated from the digest using centrifugation at 500× *g* for 10 min at 4 °C. Supernatant (trypsin-digest supernatant) was separated from the cell pellet and cells were resuspended with sterile filtered 0.9% NaCl and counted.

For the whole cell lysate, 1 mL aliquot of cell suspension was taken and re-spun at 500× *g* for 10 min at 4 °C. Supernatant was aspirated and discarded and cell pellet was resuspended with 200 µL of NP-40 lysis buffer with 1% proteinase inhibitor cocktail (whole cell pellet lysate).

The remaining cell suspension was used for plasma membrane preparation following the method of Fitzpatrick et al. [95] Briefly, cells were separated from saline by centrifugation at 500× *g* for 10 min at 4 °C. Saline supernatant was aspirated and discarded and the cell pellet was resuspended with 5 mL of 0.25 M sucrose (pH 7.4). The pellet was vortexed and homogenized in a glass, tabletop homogenizer then centrifuged at 1480× *g* for 20 min at 4 °C (3.5 K RPM in 50.2 Ti Rotor using Beckman Coulter Class R Ultracentrifuge). Supernatant was discarded and pellet was resuspended with 5 mL of 2 M sucrose and vortexed, then centrifuged at 20,000× *g* for 20 min at 4 °C (12.9 K RPM in 50.2 Ti rotor). The supernatant was transferred to a new tube and 20 mL of ice-cold ultrapure H_2_O was added to each tube. Tubes were vortexed, then centrifuged at 40,000× *g* for 30 min at 4 °C (18.2 K RPM in 50.2 Ti rotor). The resulting supernatant was discarded and final plasma membrane pellet was re-suspended in sterile-filtered 0.9% NaCl.

MV isolation using differential centrifugation has been previously described [96]. The trypsin-digest supernatant was centrifuged at 25,000× *g* for 20 min at 4 °C (16.4 K RPM in 50.2 Ti rotor) to pellet any remaining cell debris. The supernatant was transferred to a new tube and matrix vesicles were pelleted via ultracentrifugation at 100,000× *g* for 70 min at 4 °C (33 K RPM in 50.2 Ti rotor). This supernatant was carefully discarded, and the pellet was resuspended with 20 mL of sterile-filtered 0.9% NaCl for washing and ultracentrifuged again at 100,000× *g* for 70 min at 4 °C as above. The final supernatant was carefully discarded, and the pellet of matrix vesicles was resuspended in 100 µL sterile-filtered 0.9% NaCl for assays.

EX isolation using differential centrifugation has also been previously described [97]. The culture media were initially spun at 500× *g* for 10 min at 4 °C, then the supernatant was spun at 2000× *g* for 10 min at 4 °C. The supernatant was transferred to ultracentrifuge tubes and spun at 10,000× *g* for 30 min at 4 °C (10.5 K RPM in 50.2 Ti rotor). The supernatant was transferred to a new tube and exosomes were pelleted at 100,000× *g* for 70 min at 4 °C, similar to MVs above. After carefully discarding the supernatant, the pellet was resuspended with 20 mL of sterile-filtered 0.9% NaCl and ultracentrifuged again at 100,000× *g* for 70 min at 4 °C as above. The final supernatant was carefully discarded and the pellet of EXs was resuspended in 100 µL sterile-filtered 0.9% NaCl for assays. MVs and EXs were measured for their protein content using a Pierce BCA Protein Assay Kit (Thermo Fisher Scientific, Waltham, MA, USA).

### 4.4. Alkaline Phosphatase-Specific Activity

Following whole cell lysate, plasma membrane, MV, and EX isolation, each fraction was assayed for alkaline phosphatase specific activity via the formation of *para*-nitrophenol from *para*-nitrophenylphosphate at pH 10.2. Protein content for normalization was quantified using a Pierce BCA Protein Assay Kit as above.

### 4.5. Transmission Electron Microscopy

Isolated matrix vesicles were fixed in Karnovsky fixative (Electron Microscopy Sciences, Hatfield, PA, USA). A single drop of fixed MVs was placed on parafilm and a formvar-carbon-coated copper grid (Electron Microscopy Sciences, Hatfield, PA, USA) was floated in the suspension, carbon coated-side in the suspension, for 20 min. The grid was then washed in droplets of ultrapure water for 2 min per droplet for a total of 8 washes. After washing, the grid was transferred to a droplet of uranyl-oxalate solution for 5 min. Uranyl-oxalate was prepared by combining equal parts 2% Uranyl acetate (Electron Microscopy Sciences, Hatfield, PA, USA) with 0.075 M oxalic acid and adjusting the pH to 7 with 25% NH_4_OH. Finally, the grid was embedded in a droplet of 0.2% uranyl acetate and 1.8% methylcellulose for 10 min on ice. The excess liquid was removed and the grid was dried at room temperature. The grid was viewed at 50,000× and 100,000× magnification and 200 kV using an electron microscope (JEM-F200 Cold FEG Electron Microscope, Jeol, Peabody, MA, USA).

### 4.6. Nanoparticle Tracking Analysis

The size distributions of the matrix vesicles and exosomes were determined via nanoparticle tracking analysis. A volume of 10 µL of vesicle suspension was mixed with 10 µL of 0.02% paraformaldehyde in phosphate buffered saline (PBS). MVs and EXs were then diluted 10 and 5 times, respectively, in paraformaldehyde-PBS solution before completing 3 repeats of nanoparticle tracking analysis using a Zetaview (Analytik, Cambridge, UK) per the manufacturer’s instructions.

### 4.7. Western Blot

From each cell pellet lysate, plasma membrane, MV, and EX fraction, a total of 3 µg of protein were subjected to 4–20% mini-Protean TGX gels (Bio-Rad, Hercules, CA, USA) and transferred to a PVFD membrane (Bio-Rad). After 1 h of blocking in 5% nonfat milk in PBS containing 0.1% Tween-20 (PBS-T) at room temperature, membranes were incubated overnight with antibodies. Membranes were washed in PBS-T and incubated with species-specific horseradish peroxidase-labeled secondary antibody for 1 h at room temperature. After a second washing with PBS-T, a Pierce ECL chemiluminescence system (Thermo Fisher Scientific, Waltham, MA, USA) was used to develop membranes to detect bound antibodies. GAPDH served as the internal control for the CP and PM fractions, while ALIX and HSP70 served as internal controls for MVs and EXs which do not express GAPDH.

### 4.8. PKH26 Labeling and Confocal Microscopy

Purified MVs were labeled with PKH26 red fluorescent cell membrane label (Sigma-Aldrich, St. Louis, MO, USA). Briefly, the PKH26 dye in manufacturer’s diluent was mixed with MV suspension or sterile 0.9% sodium chloride (negative control) and pipetted to mix once per minute for five minutes. After 5 min, 1 mL of sterile 1% bovine serum albumin was added to each tube to quench unbound dye. A total of 8 mL of basal medium was added to each tube, tubes were balanced, and ultracentrifuged at 100,000× *g* for 70 min at 4 °C (33 K RPM in 50.2 Ti r). The supernatant was carefully removed, and the pellet was resuspended in 100 mL of sterile DPBS. A total of 8 × 10^3^ cells were plated in each well of a CCS-8 well culture slide. MVs (0.85 µg protein) were added to the antibiotic-free culture media when cells reached 70% confluence. After 24 h of incubation, culture media were removed, and cells were washed twice with sterile DPBS and then fixed with 10% neutral buffered formalin. The nuclei (DRAQ5, Abcam, Cambridge, UK) and plasma membrane (CellBrite Green, Biotium, Fremont, CA, USA) were stained according to manufacturer’s instructions. Twelve random fields of view from each group were imaged using a confocal microscope (LSM 980, Zeiss, Oberkochen, Germany). Two independent observers reviewed all 24 images for total cell number and total number of cells with at least one distinct spot of PKH26 fluorescence.

### 4.9. microRNA Extraction and Detection

microRNAs from MG63 MVs and whole cells were extracted using QIAzol (Qiagen, Hilden, Germany) and the miRNeasy micro extraction kit (Qiagen, Hilden, Germany) as per the manufacturer’s instructions. Briefly, QIAzol lysis reagent homogenized each sample. Chloroform addition followed by centrifugation separated the homogenate into aqueous and organic phases. The aqueous phase with the RNA was isolated from the DNA interphase and protein organic phase. Isopropanol precipitated the RNA from the aqueous phase with miRNeasy spin column collection. Using the spin column, the RNA was washed with ethanol. Finally, RNA was eluted in RNase-free water and RNA quantity and quality were assessed via bioanalyzer analysis with an RNA 6000 Pico kit (Agilent Technologies, Santa Clara, CA, USA) and an Agilent 2100 Bioanalyzer per manufacturer’s instructions (Agilent Technologies). Eighteen independent MV samples were pooled into three samples for sequencing with at least 750 ng of RNA per sample. Corresponding cell samples were pooled in the same manner for a total volume of 60 µL.

### 4.10. microRNA Sequencing (RNA-Seq)

Following verification of RNA quality and quantity, samples were sent to GENEWIZ for next-generation sequencing (Azenta Life Sciences, Burlington, MA, USA). Library preparation was completed using size fractionation with adapter ligation to the 5′ phosphate of the microRNA. Libraries were pooled and sequenced 2 × 150 base pair (bp) on the Illumina HiSeq 2500. Raw RNA-seq reads (FASTQ) were aligned to the respective H. sapiens genome using miARma-seq. The read count data were then normalized, and statistical tests were performed to determine differentially expressed genes (adjusted *p* value using Benjamin-Hochberg method, <0.05 and fold change ≥ 2) using DESeq2.

### 4.11. Pathway Analysis of microRNA Associations

microRNAs were selected from the differential expression list using the Log2 fold change (>2) and the adjusted *p*-value (<0.05). Those with the greatest Log2 fold change were searched within PubMed for their miRNA along with the terms: “osteoblast”, “bone”, or “osteogenesis” to begin the literature review. Peer-reviewed publications were selected based on title and abstract, then further reviewed for association between selected miRNA, osteogenic potential, and macrophage polarization. The resulting classifications were based on at least one publication or deemed inconclusive if two or more publications were found with conflicting results.

### 4.12. MV and EX Treatment and Osteogenic Response

MG63 osteoblast-like cells were cultured as described. After reaching confluence, culture media were replaced with fresh DMEM with 10% FBS without pen-strep and cells were treated with 5 µg/mL (based on protein content) of MVs (either GM or OM as described above) or EXs for 24 or 48 h. At the time of harvest, conditioned media were collected, cells were washed with 1 mL of phosphate-buffered saline (PBS), and lysed with 500 µL 0.05% Triton X-100 in PBS per well. After a single freeze-thaw, cells were sonicated. From the cell lysate, DNA quantification and alkaline phosphatase-specific activity were assessed using Quantifluor dsDNA system (Promega, Madison, WI, USA) and via the formation of para-nitrophenol from para-nitrophenylphosphate at pH 10.2, respectively. Osteocalcin (R&D, Minneapolis, MN, USA), osteopontin (R&D Systems, Minneapolis, MN, USA), vascular endothelial growth factor (R&D Systems), interleukin-6 (Peprotech, Cranbury, NJ, USA), and interleukin-10 (Peprotech, Cranbury, NJ, USA) levels were measured from the conditioned media using enzyme-linked immunoassay (ELISA) following the manufacturer’s instructions.

### 4.13. Gene Expression

MG63 osteoblast-like cells were cultured and treated with 5 μg/mL MVs (GM) as described above. At the time of harvest, culture media were aspirated and RNA was extracted uding TriZol (Invitrogen, Waltham, MA, USA). RNA was quantified using the Take3 Microvolume Plate (Biotek, Winooski, VT, USA) and used to synthesize cDNA libraries. The expression of BGLAP (OCN), RUNX2, and Osterix/SP7 were quantified by real-time quantitative PCR using Power SybrGreen^®^ Master Mix (Applied Biosystems, Waltham, MA, USA) and gene-specific primers from Thermo Fisher Scientific (Table 3). The expression levels of each gene were normalized to the levels of glyceraldehyde-3-phosphate dehydrogenase (GAPDH).

### 4.14. Methyl-β-Cyclodextrin Endocytosis Inhibition Treatment

MG63 osteoblast-like cells were cultured as described above in 24-well plates. After reaching confluence, culture media were replaced with fresh basal DMEM supplemented with methyl-β-cyclodextrin (β-CD; Thermo Fisher Scientific, Waltham, MA, USA). Briefly, stock 100 mM β-CD was prepared in sterile ultrapure H_2_O. The stock solution was then serially diluted into 10 mM, 1 mM, and 0.1 mM β-CD in basal DMEM and added to cells for 30 min, based on previous studies demonstrating that this incubation time destroyed caveolae but was not cytotoxic [98]. A vehicle of ultrapure H_2_O at 10 mM was also prepared as a control. After 30 min, cells were washed twice with basal DMEM followed by treatment with 5 µg protein/mL MVs or an equivalent amount of sterile 0.9% NaCl in DMEM with 10% FBS and without pen-strep for 48 h. At the end of MV treatment, cells were harvested for conditioned media and cell lysate in 0.05% Triton X-100 in PBS, as described above. DNA quantification and alkaline phosphatase-specific activity were assessed using a Quantifluor dsDNA system (Promega, Madison, WI, USA) and via the formation of *para*-nitrophenol from *para*-nitrophenylphosphate at pH 10.2, respectively.

### 4.15. Statistical Analysis

The results of the culture study are presented as mean ± standard error for six independent cultures per variable. Between-group differences with only two groups were calculated using an unpaired *t*-test, while between-group differences with three or more groups were calculated using one-way ANOVA with a post-hoc Tukey HSD. A *p*-value less than 0.05 was considered significant. All statistically significant *t*-tests are denoted by * and all statistically significant ANOVA analyses are denoted by groups not sharing a letter. Experimental observations were validated with at least one experimental repeat. Statistical analysis and presentation for all experiments were carried out using GraphPad Prism 9.3.1 and JMP Pro 16.

## Figures and Tables

**Figure 1 ijms-24-12770-f001:**
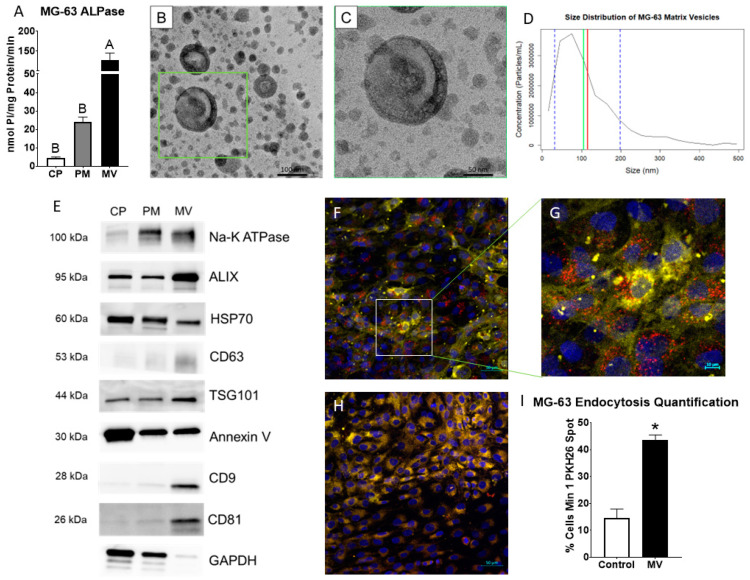
**Matrix vesicles display extracellular vesicle characteristics and demonstrate uptake by osteoblast-like cells.** Quantification of alkaline phosphatase specific activity for cell pellet lysate (CP), plasma membrane (PM), and matrix vesicle (MV) isolates (**A**); transmission electron microscopy at 50 K magnification of MV demonstrated circular vesicles with a bilaminar membrane ((**B**), scale bar = 100 nm); TEM of green inset in B at 100 K magnification ((**C**), scale bar = 50 nm); size distribution of osteoblast-derived MVs using nanoparticle tracking analysis with median of 105 nm (green line), mean of 115 nm (red line) and standard deviation of ±85 nm (blue dashed lines); the results are mean of three independent measurements (**D**); Western blot analysis of protein expression in CP, PM, and MV isolates (**E**); osteoblasts treated with PKH26-stained MVs (red fluorescence), DRAQ5 nuclear counterstain (blue fluorescence) and plasma membrane counterstain (yellow fluorescence) using confocal microscopy ((**F**), scale bar = 50 µm); region of interest to highlight PKH26-stained MVs surrounding osteoblast nuclei ((**G**), scale bar = 10 µm); NaCl control-treated osteoblasts with DRAQ5 nuclear counterstain and plasma membrane counterstain ((**H**), scale bar = 50 µm); quantification of cells with a minimum of one spot of MV fluorescence between MV and NaCl control images (**I**). Results are expressed as the mean ± SEM of 2 independent reviewers. Groups not sharing a letter are considered statistically different at α = 0.05 by one-way ANOVA with Tukey post hoc test. Group with * is significantly different from the vehicle at an α = 0.05 using unpaired *t*-test.

**Figure 2 ijms-24-12770-f002:**
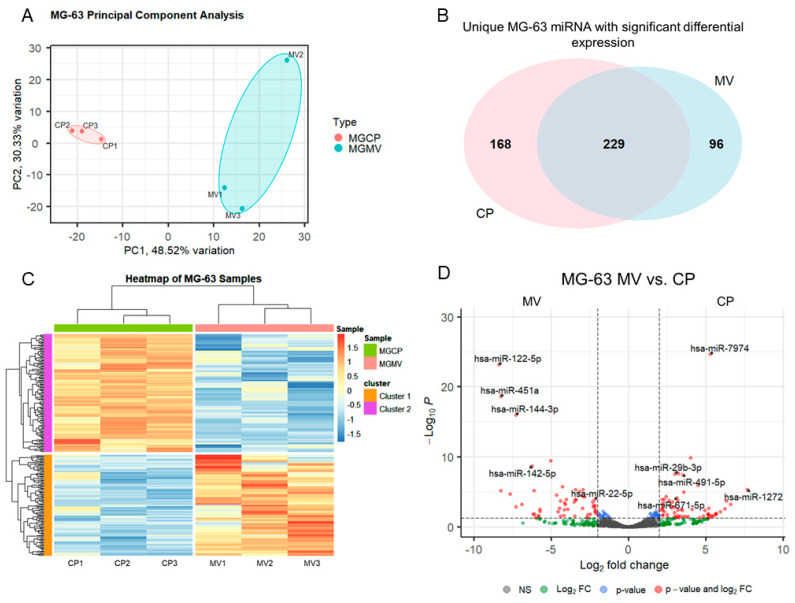
**Small RNA sequencing suggests selective packaging of microRNA into MG63 matrix vesicles.** Principal component analysis plot of components 1 and 2 for all six samples with grouping of cell lysate (CP) samples (pink) and matrix vesicle (MV) samples (teal) (**A**); Venn diagram displaying overlap of unique microRNA with differential expression between CP and MV samples (**B**); heatmap of sample z-scores clustered with Euclidean distance measure (**C**); volcano plot of microRNA with Log2 fold change greater than ±2 and adjusted *p*-value less than 0.05 (**D**).

**Figure 3 ijms-24-12770-f003:**
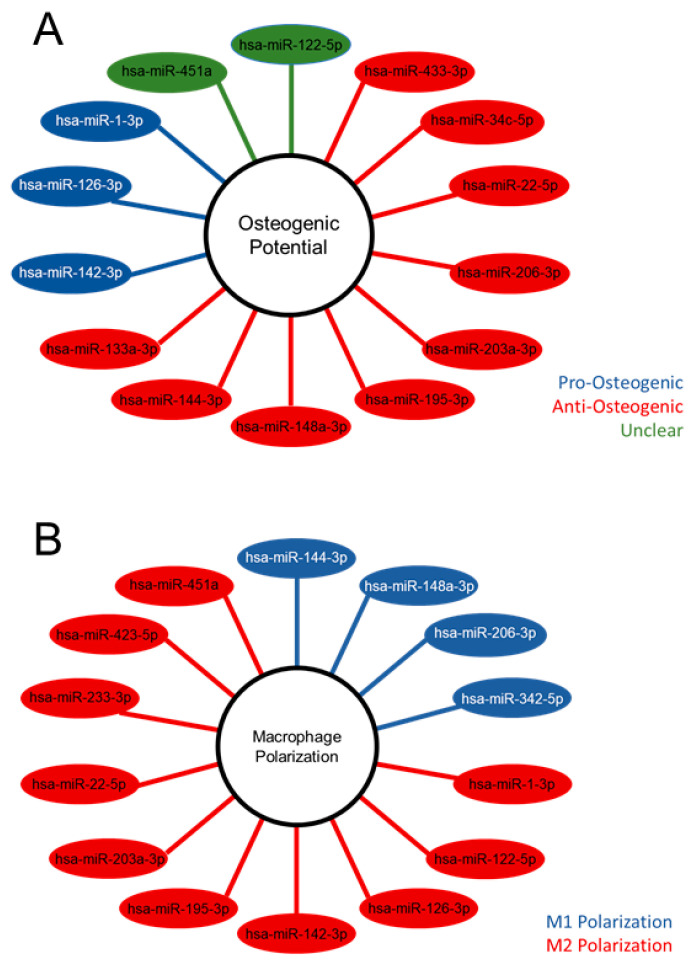
Differentially expressed microRNA of matrix vesicles suggest an anti-osteogenic role in osteoblasts and regenerative role in macrophages. Differentially expressed microRNA (adjusted *p*-value < 0.05 and absolute Log2 fold change >2) with pro-, anti-, or unclear osteogenic potential based on the literature search (**A**); differentially expressed microRNA with promotion of M1 or M2 macrophage polarization according to literature search (**B**).

**Figure 4 ijms-24-12770-f004:**
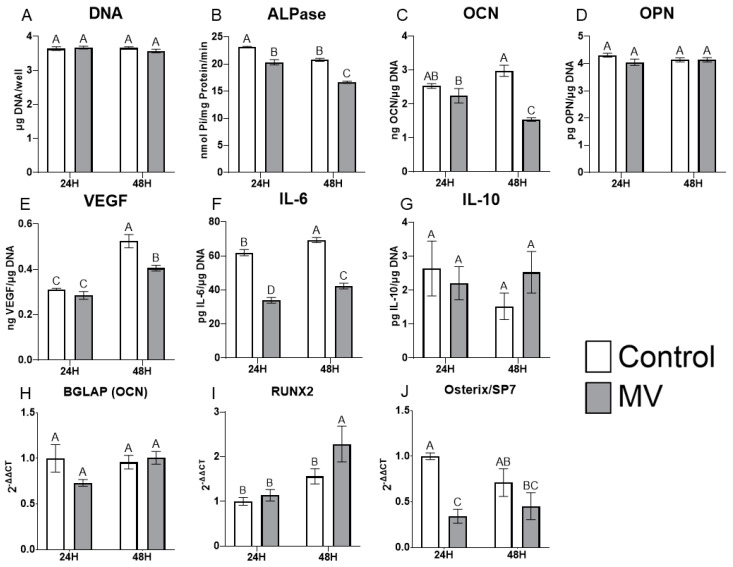
**Treatment of MG63 cells with matrix vesicles decreases markers of differentiation.** Quantification of DNA (**A**), alkaline phosphatase (**B**), osteocalcin (**C**), osteopontin (**D**), vascular endothelial growth factor (**E**), interleukin-6 (**F**), and interleukin-10 (**G**) in MG63 cells treated for 24 or 48 h with or without 5 µg/mL matrix vesicles (MV) harvested from MG63s grown in growth media until 24 h after confluence; relative quantification of messenger RNA (mRNA) expression of BGLAP (**H**), RUNX2 (**I**), and Osterix/SP7 (**J**) in MG63 cells treated for 24 or 48 h with or without 5 µg/mL matrix vesicles (MV) from the same conditions. mRNA was quantified via RT-PCR and normalized to GAPDH expression. Data are from a representative experiment and are shown as mean ± SEM of n = 6 per group. Groups not sharing a letter are considered statistically different at α = 0.05 using one-way ANOVA with Tukey post-hoc test.

**Figure 5 ijms-24-12770-f005:**
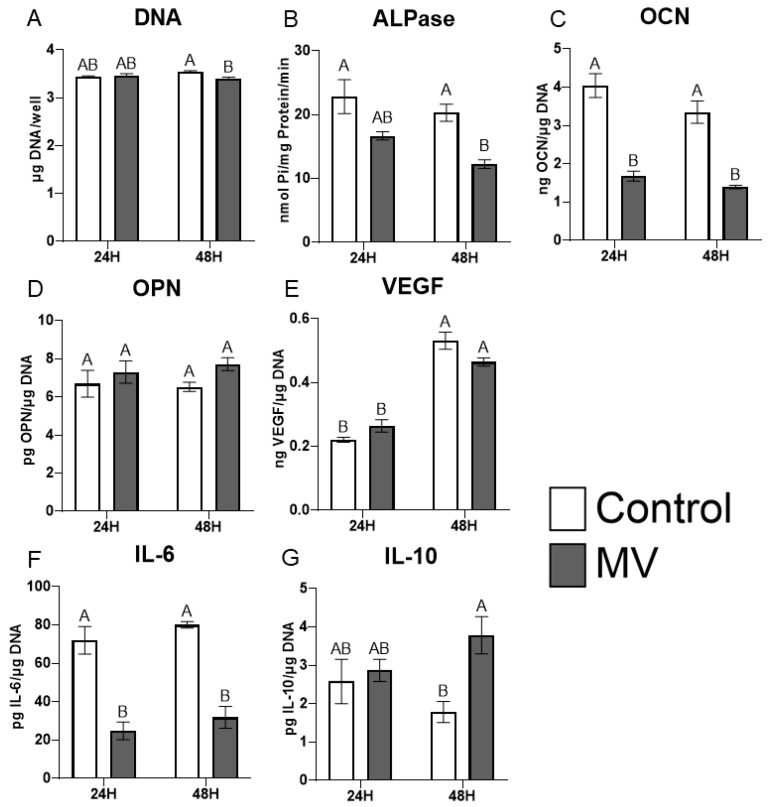
**Matrix vesicle composition is sensitive to culture conditions.** Quantification of DNA (**A**), alkaline phosphatase (**B**), osteocalcin (**C**), osteopontin (**D**), vascular endothelial growth factor (**E**), interleukin-6 (**F**), and interleukin-10 (**G**) in MG63 cells treated for 24 or 48 h with or without 5 µg/mL matrix vesicles (MV) harvested from MG63s grown in osteogenic media for 10 days. Data are from a representative experiment and are shown as mean ± SEM of n = 6 per group. Groups not sharing a letter are considered statistically different at α = 0.05 using one-way ANOVA with Tukey post-hoc test.

**Figure 6 ijms-24-12770-f006:**
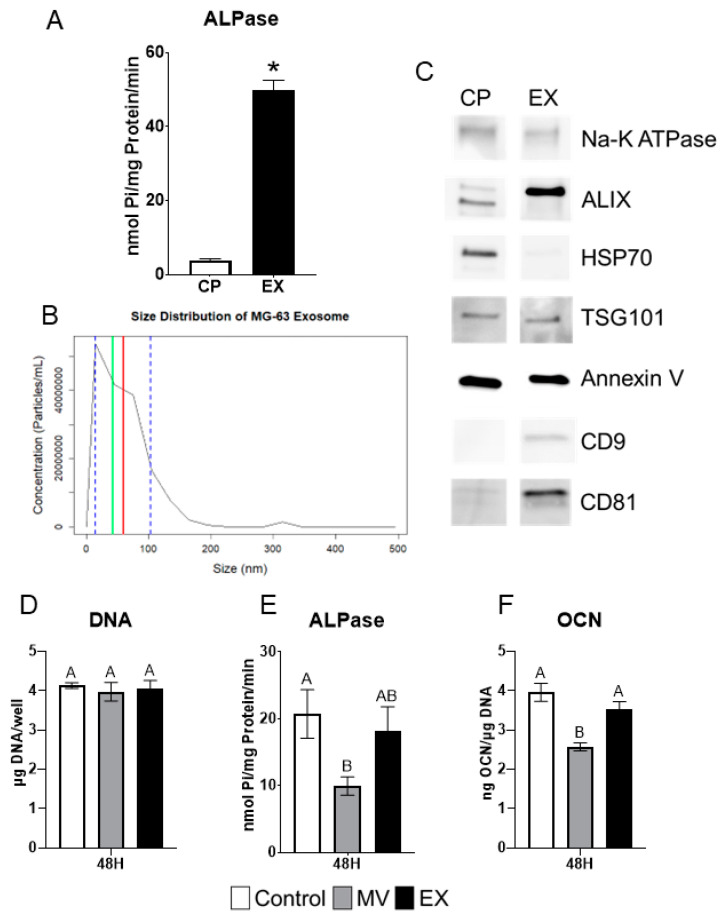
**MG63 exosomes display characteristics and treatment effects different from matrix vesicles.** Quantification of alkaline phosphatase specific activity for cell pellet lysate (CP) and exosome (EX) isolates (**A**); Size distribution of MG63-derived EXs using nanoparticle tracking analysis with median of 42 nm (green line), mean of 59 nm (red line) and standard deviation of ±44.6 nm (blue dashed lines), results are mean of three independent measurements (**B**); Western blot analysis of protein expression in CP and EX isolates (**C**); Quantification of DNA (**D**), alkaline phosphatase (**E**), and osteocalcin (**F**) in MG63 cells treated for 24 or 48 h with 5 µg/mL matrix vesicles (MV), EX, or saline control. MVs and EXs were harvested from the trypsinized cell layer or conditioned media, respectively, of MG63s grown in growth medium until 24 h after confluence. Data are from a representative experiment and are shown as mean ± SEM of n = 6 per group. Group with * is significantly different from the vehicle at an α = 0.05 by unpaired *t*-test. Groups not sharing a letter are considered statistically different at α = 0.05 using one-way ANOVA with Tukey post-hoc test.

**Figure 7 ijms-24-12770-f007:**
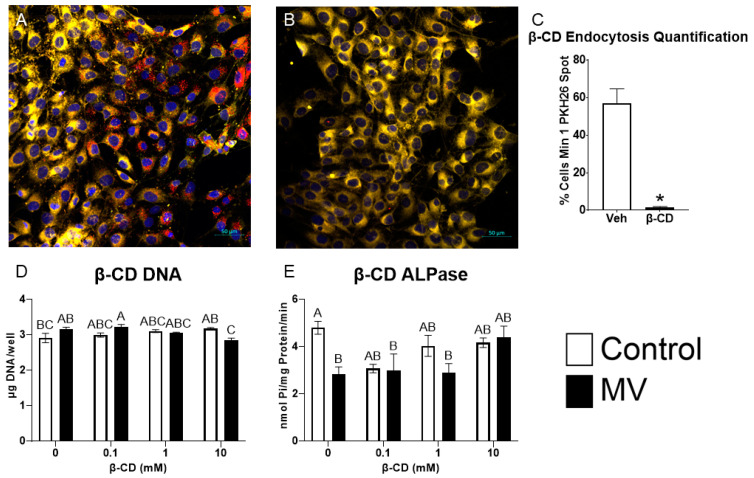
**Inhibition of caveolae-mediated endocytosis prevents uptake and effect of matrix vesicles on osteoblast-like cells.** MG63 cells pre-treated with 10 mM vehicle (**A**) or methyl-β-cyclodextrin (β-CD; (**B**)) followed by treatment with PKH26-stained MVs (red fluorescence), DRAQ5 nuclear counterstain (blue fluorescence) and plasma membrane counterstain (yellow fluorescence) using confocal microscopy (scale bar = 50 µm); quantification of cells with a minimum of one spot of MV fluorescence between vehicle and β-CD images (**C**); quantification of DNA (**D**) and alkaline phosphatase-specific activity (**E**) in MG63 cells pre-treated with increasing concentrations of β-CD followed by treatment for 48 h with or without 5 µg/mL matrix vesicles (MV) harvested from MG63s grown in growth media until 24 h after confluence. Results are expressed as the mean ± SEM of 2 independent reviewers. Group with * is significantly different from the vehicle at an α = 0.05 by unpaired *t*-test. Groups not sharing a letter are considered statistically different at α = 0.05 by one-way ANOVA with Tukey post-hoc test.

**Figure 8 ijms-24-12770-f008:**
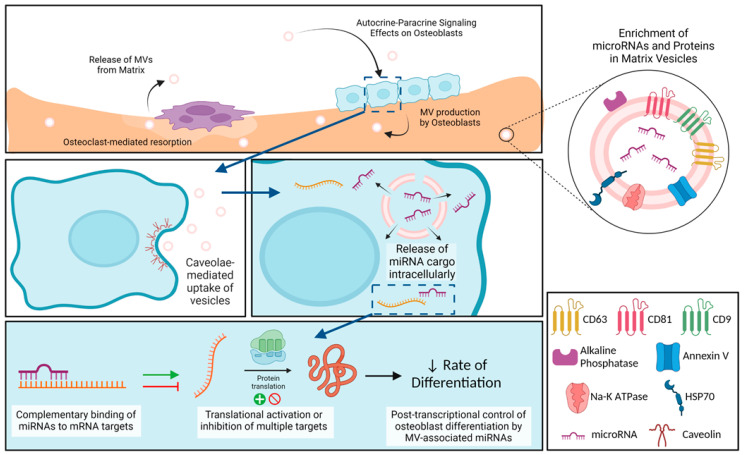
**Schematic representation of osteoblast-derived matrix vesicles.** Matrix vesicles (MVs), with their protein and microRNA (miRNA) enrichment, are produced by osteoblasts and become embedded in the mineralized extracellular matrix. During osteoclast-mediated resorption of bone, MVs are released from the mineralized matrix to act in an autocrine and/or paracrine fashion on nearby osteoblasts. These MVs are taken up by osteoblasts via caveolin-mediated endocytosis, where they are able to release their miRNA cargo intracellularly. miRNAs interact with target messenger RNA (mRNA) to active or inhibit translation into proteins, slowing production of osteoblastic differentiation markers. Image created with BioRender.com.

**Table 1 ijms-24-12770-t001:** Differentially Expressed microRNA and Osteogenic Potential.

miRNA	Osteogenic Potential	References
hsa-miR-1-3p	Pro-Osteogenic	[24]
hsa-miR-126-3p	Pro-Osteogenic	[25,26]
hsa-miR-142-5p	Pro-Osteogenic	[27,28,29]
hsa-miR-133a-3p	Anti-Osteogenic	[30,31,32]
hsa-miR-144-3p	Anti-Osteogenic	[33,34,35,36]
hsa-miR-148a-3p	Anti-Osteogenic	[26,37,38]
hsa-miR-195-3p	Anti-Osteogenic	[39]
hsa-miR-203a-3p	Anti-Osteogenic	[40]
hsa-miR-206	Anti-Osteogenic	[41,42,43,44]
hsa-miR-22-5p	Anti-Osteogenic	[45,46]
hsa-miR-34c-3p	Anti-Osteogenic	[47]
hsa-miR-433-3p	Anti-Osteogenic	[48,49]
hsa-miR-122-5p	Unclear	[14,50]
hsa-miR-451a	Unclear	[51,52]

**Table 2 ijms-24-12770-t002:** Differentially Expressed microRNA and Inflammatory Modulation.

miRNA	M1/M2 Polarization	References
hsa-miR-144-3p	M1	[53,54]
hsa-miR-148a-3p	M1	[55,56]
hsa-miR-342-5p	M1	[57]
hsa-miR-206-3p	M1	[58]
hsa-miR-1-3p	M2b	[59]
hsa-miR-122-5p	M2	[60]
hsa-miR-126-3p	M2	[61,62]
hsa-miR-142-5p	M2	[63,64,65]
hsa-miR-195-3p	M2	[66,67]
hsa-miR-22-5p	M2	[68]
hsa-miR-223-3p	M2	[65]
hsa-miR-328-5p	M2	[69]
hsa-miR-423-5p	M2	[70]
hsa-miR-451a	M2	[71,72]

**Table 3 ijms-24-12770-t003:** Gene-specific Primers.

Gene	Forward	Reverse
BGLAP	CGCTACCTGTATCAATGGCTGG	CTCCTGAAAGCCGATGTGGTCA
RUNX2	CCCAGTATGAGAGTAGGTGTCC	GGGTAAGACTGGTCATAGGACC
Osterix/SP7	TTCTGCGGCAAGAGGTTCACTC	GTGTTTGCTCAGGTGGTCGCTT
GAPDH	GCTCTCCAGAACATCATCC	TGCTTCACCACCTTCTTG

## Data Availability

All data are available in the manuscript. Additional information will be provided following receipt of reasonable written request.

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
