# Peer review of "Osteoblast-Derived Matrix Vesicles Exhibit Exosomal Traits and a Unique Subset of microRNA: Their Caveolae-Dependent Endocytosis Results in Reduced Osteogenic Differentiation"

_ijms, 2023, doi:10.3390/ijms241612770_

Round 1

Reviewer 1 Report

This manuscript is interesting but, in my opinion, it has some limitations.

Different articles, even very dated, indicate the similarity of the MG63 cell line with normal osteoblasts. Indeed, they respond in a very similar way to osteoblasts to the administration of active vitamin D (1,25(OH)2 D3) and PTH, which makes them an ideal model for studies of hormone regulation and phenotypic changes (Kumarasuriyar et al. 2009). However, the inconsistency in the expression of matrix proteins already limits their use as a model for the development of this phenotype and on matrix mineralization, as this cell line remain almost undifferentiated, and do not mineralize the matrix  (Pierschbacher et al., 1988; Saldana et al., 2011). Therefore, the fact that matrix microvesicles isolated from MG63 inhibit osteoblastic differentiation comes as no surprise to me. Microvesicles, despite having some factors selectively accumulated inside the vesicle, often have a content similar to that of the cells that produce them, so if these cells maintain an undifferentiated state from epigenetic regulatory mechanisms (miRNAs, cirRNA, lncRNA, etc.), it is clear that this influence also affects exosomes/microvesicles produced by them.

To confirm what they claim the authors, i.e. that exosomes/microvesicles of osteoblasts inhibit osteoblastic differentiation, should have used exosomes/Microvesicles isolated from normal osteoblasts (primary Ob cells),  and not by osteosarcoma cell line.

the manuscript requires only minor stylistic changes.

Reviewer 2 Report

In this manuscript, the authors indicate the enrichment of miRNA within osteoblast-derived MVs, in which miRNA can inhibit osteoblast differentiation and promote a regenerative inflammatory state. Therefore, the authors suggest that osteoblast-derived MVs may modulate the rate of osteogenesis as well as contribute in a much more nuanced manner involving M2 polarization of macrophages to a regenerative phenotype.

This manuscript is reasonable and organized well. However, I think there are some questions as follows,

Major comments

1. Figure 1E: Please add the expression data of each internal control.

2. Figure 4/5: As for osteogenesis, I think the authors should investigate the longer time point (for example 1 week or 2 weeks). Moreover, to refer to osteogenesis, the authors should investigate the master regulatory osteogenesis-related genes, for example, Runx2 or Osterix. Please add these data.

Minor comments

1.       The authors indicate characteristic miRNAs in osteoblast-derived MVs and conclude these miRNAs can regulate osteogenesis and regenerative inflammation. However, we are interested in what these key regulatory factors are. I would like the authors to mention the candidate of the regulatory miRNA. Please mention this question with some previous reports or additional data in the DISCUSSION section.

Reviewer 3 Report

This manuscript explores the role of MVs in bone meneralisation and describes their characteristics using different approaches. The authors report that MVs have increased ALP activity in comparison with plasma membranes but are enriched in antiosteogenic miRNAs. Interestingly, these DE miRNAs play a role in M2 macrophage polarization.

The work is interesting and provides useful information for this specific topic. However, the authors need to address some points:

1.       Why the authors used MG63 cells and not a cell line without cancer cells phenotype? Although MG63 has been widely used as an osteoblast-like cell model, these cells still have altered characteristics as compared to normal osteoblasts.

2.       Tables 1 and 2 and Figure 3 provide the same information for the role of miRNAs. It is better to keep one of them, not both.

3.       Do hsa-miR-122-5p, hsa-miR-451a, and hsa-miR-144-3p have any osteoblast/bone specific targets? These should be also assessed to unravel their role in osteogenesis/mineralisation.

4.       It would be very informative to assess terminal mineralisation by bone nodule formation and Alizarin Red S or von Kossa staining.

Round 2

Reviewer 2 Report

 I checked these answers and revisions, and I have agreed with all your answers and revisions.

I thank the authors for your answer and revision.

I think this revised manuscript can be acceptable.